# Injury Incidence, Outcomes, and Return to Competition Times after Sports-Related Concussions during One Professional Ice Hockey Season: A Prospective Cohort Study

**DOI:** 10.3390/healthcare11243153

**Published:** 2023-12-12

**Authors:** Dominik Höllerer, Peter Kaiser, Armin Runer, Ekkehard Steiner, Christian Koidl, Rohit Arora, Friedemann Schneider

**Affiliations:** 1Faculty of Medicine, Medical University of Innsbruck, 6020 Innsbruck, Austria; dominik.hoellerer@student.i-med.ac.at; 2Department of Orthopaedics and Traumatology, Medical University of Innsbruck, Anichstraße 35, 6020 Innsbruck, Austria; 3HC TIWAG Innsbruck, 6020 Innsbruck, Austria; 4Department of Sports Orthopaedics, Technical University of Munich, 81675 Munich, Germany

**Keywords:** adult athletes, concussion, head impact, traumatic brain injuries, ice hockey, injury epidemiology

## Abstract

Objective: The objective of this study was to analyze the incidence and characteristics of sports-related concussions (SRCs) for a professional ice hockey team during one regular season in the International Central European Hockey League. Background: Repeated concussions are a common cause of long periods of absence in team contact sports, with a wide range of potential short- and long-term consequences for the affected athlete. Questions mainly regarding early diagnosis and ideal follow-up treatment remain unanswered, especially regarding the timing of return to sports (RTS). Method: A prospective data analysis for a professional ice hockey team during a regular season was conducted. Firstly, concussions per 1000 athlete exposure (AE) and average time loss due to SRC were calculated. Secondly, the data from ImPACT Applications were analyzed for those players who were diagnosed with an SRC. Results: Five SRCs were evaluated during the regular season, which resulted in 1.35 concussions/1000 AEs, a maximum of 17 missed games, and a median of three games per SRC. The average symptom count was 9.6, with the most common symptoms being headache, sensitivity to light, and dizziness. Conclusions: SRCs sustained in professional ice hockey are a common in-competition injury, while practices play a subsidiary role. The duration of RTS is highly individual and can be associated with symptoms lasting days to months.

## 1. Introduction

Concussions, sustained during recreational and professional sports activity, can result in severe short- and long-term impairments [1,2]. A recent study from 2020 showed that in Ontario, Canada, on average, 1153/100,000 people were diagnosed with a concussion each year [3]. Sports-related concussions (SRCs) are a common injury in competition, especially among ice hockey players [4,5,6,7]. For example, in the NHL, 5.8 to 6.1 concussions occur per 100 games [8].

Regarding diagnostics and the consequential treatment of concussions in general and SRCs in particular, there are some relevant difficulties. First, there is still no conclusive, uniform definition of a concussion. The most appropriate definition in the context of sport comes from the Concussion in Sport Group, which defines a sports-related concussion as a traumatic brain injury caused by a direct blow to the head, neck, or body that subsequently triggers a complex neurotransmitter and metabolic cascade, with possible axonal injury, blood flow change, and inflammation affecting the brain. Corresponding symptoms can occur immediately or with a delay and usually disappear within a few days to weeks, but sometimes only after a longer period of time [9].

Second, over the course of the last few years, many studies have found a relationship between SRC and a higher risk of sustaining lower extremity musculoskeletal injuries after a return to sports (RTS) [2,10,11].

Recognizing a concussion and seeking appropriate treatment immediately after a head injury is essential to prevent further damage and to ensure optimal recovery.

The current literature emphasizes that “there have been few European studies on ice hockey injuries in the past two decades” [2,10,11]. Nevertheless, the necessity is no less in Europe, though the conditions of playing ice hockey are considerably divergent. One of these dissimilarities is the size of the playing field, as European rinks (according to the International Ice Hockey Federation) are larger than those in North America (in accordance with the requirements of the National Hockey League) [12]. In addition, fights are generally penalized more severely in Europe than in North American leagues and are therefore not as prevalent as in North American leagues [13].

Many studies on ice hockey injuries have focused on children, adolescents, or college sports. Overall, the recent literature concerning adult professional ice hockey in Europe is sparse [6,14,15,16].

The primary aim of this study was to determine the number of SRCs per 1000 athlete exposures (AEs) diagnosed during one regular season of play in the International Central European Hockey League (ICE Hockey League). Furthermore, the severity and impact of SRC on professional ice hockey players as well as the whole team were investigated.

## 2. Materials and Methods

### 2.1. Design, Setting, and Definition

A prospective study of one professional ice hockey team playing in the ICE Hockey League was conducted. Every practice and game during the regular season (including the preseason) lasting from August 2022 to February 2023 was considered, which added up to a total of 29 weeks. League rules required a trained physician to be present at every game. In addition, every player who was on the roster at the beginning of the season or acquired during the season had to take a baseline concussion test before their first regular game.

This baseline test was administered via ImPACT Applications (ImPACT Applications, Inc., Coralville, IA, USA, 2023, product version 4.0) during the preseason period in September 2022, consisting of self-reported symptoms and a neurocognitive (NC) test battery. This battery evaluates attentional processes, verbal recognition memory, visual recognition memory, visual working memory, visual processing, motor speed, and reaction time.

SRCs were diagnosed by the responsible team physicians. In case of a suspected SRC, the player was removed from the game or practice and not allowed to rejoin activities the same day. If possible, within the next 24 h, an NC test was carried out and self-reported symptoms were queried via the ImPACT tool. These test results helped the team physicians to confirm or reassess the situation regarding suspected SRC.

If players reported symptoms suggestive of concussion at a later time, testing was performed within 24 h of self-reported concern. This was the case when athletes showed no obvious signs of SRC during the activity but reported concerns with a time delay after the activity due to onset or persistence of symptoms. The symptom score was in no way relied upon exclusively, but rather formed a decidedly integral part of the overall assessment, which primarily consisted of a thorough clinical examination (by the team physicians and partly by professionals from other medical disciplines), in some cases using radiological imaging and the aforementioned ImPACT test.

While the ImPACT tool was used to support team physicians’ decisions regarding SRC, the ImPACT assessment was not decisive in the diagnosis of SRC.

After the initial diagnosis of SRC, the player was withdrawn from every practice and all games until further notice. Furthermore, rest and ease were advised for the first 48 h.

The RTS process was set up by the medical team and structured with a gradual increase in load. Athletes who reported no symptoms at rest were permitted to start with step one, being aerobic exercise. If no symptoms occurred during the aerobic exercise (e.g., on the bicycle ergometer), step two, independent training on ice (e.g., skating), was started; if this was also possible without any symptoms, a return to team training without contact could be started. In step four, if the athlete tolerated this without symptoms, participation in regular training was permitted. If the athlete was still symptom-free, they were cleared to return to competition after a further medical examination. Each step lasted at least one day, which meant that the athletes could take a maximum of only one step per day. If symptoms occurred or pre-existing symptoms worsened during any of these stages, athletes were required to return to the previous stage.

### 2.2. Data Sources

In the case of injuries, further clarification was carried out in accordance with ICE Hockey League guidelines and recommendations. In the case of a clinically suspected SRC, ImPACT testing of the injured player was initiated after thorough examination by the team physicians. The study team reviewed the resulting data in comparison with the associated baseline test of the injured player. An additional written informed-consent declaration was obtained from all athletes diagnosed with SRC for further analysis and anonymized publication of the corresponding injury data. Players who finished the RTS were cleared for return to play. Players’ time loss due to injury was tracked and documented by the team physicians.

### 2.3. Study Population

The study population was the whole roster of a professional ice hockey team in the ICE Hockey League. The roster consisted of 26 players, three goalkeepers, eight defenseman, and 15 forwards. Athletes who played at least one game for the respective team during the regular 2022/2023 season were included in the study. This applied to 25 players on that roster. One player did not meet these criteria, and his data were therefore excluded from this study. Twelve athletes (48.0%) stated that they had previously suffered a concussion, while none of them reported any concussions in the previous season (2021/2022) or in the run-up to the observed season (2022/2023).

### 2.4. Statistical Analysis

Statistical analysis was carried out with Microsoft^®^ Excel^®^ for Microsoft 365 MSO (Version 2301) and IBM SPSS Statistics (Version 29.0.0.0). Data were calculated in athlete exposures (AEs) per game, per practice, and in accumulation during the whole season. An AE is one athlete participating in either one practice or one game in which they were exposed to the possibility of injury. Time practiced or played did not matter. Data for post-concussion testing and demographics were analyzed in a descriptive manner. A correlation between initial symptom score and games missed due to SRC was conducted. Two-factor analysis and norm values from ImPACT were considered.

## 3. Results

### 3.1. Incidence of Concussions

During the regular season, reaching from September 2022 to February 2023, 50 games took place, resulting in 961 AEs. Additionally, 2735 AEs were recorded during preseason and in-season practice, resulting in a cumulative 3696 AEs. Twenty-five players played at least one game for the respective team. The demographics are listed in Table 1.

During this period, five SRCs were diagnosed, all of which occurred during matches (in-competition), while no SRCs occurred during training. These figures result in a cumulative concussion risk of 1.35/1000 AEs. Details are given in Table 2. For athletes diagnosed with SRC, the median number of games missed was three; however, the maximum (max.) was 17. The median number of days missed before returning to training was 15 (min. 9, max. 50). Return to competition was, on average, one day longer, with a median of 16 days (min. 10, max. 51). Compared with other reasons for games missed, e.g., illness or other injury, concussions were substantial and the main reason for games missed (29) during the season (Table 3). Four of the five athletes (80.0%) had already suffered a concussion before. In this cohort, no athlete obtained a subsequent injury or concussion after being diagnosed with SRC.

### 3.2. Symptoms

All five concussed athletes expressed headache as the main symptom, while only 1 out of 25 players reported headache at baseline testing. Furthermore, four out of five concussed players reported dizziness and sensitivity to light. Details of the clinical symptoms and changes to the test scores are shown in Table 4 and Table 5. No apparent risk factor for prolonged recovery was found in the observed cohort.

Neurocognitive testing using the ImPACT Application showed relevant abnormalities in only two of the five players who were diagnosed with a concussion. Both players had problems with visual memory function. Their composite value from baseline to post-concussion testing declined from 90 to 61 points and 78 to 57 points, respectively, in this particular category. This also reflected the two-factor analysis from the ImPACT tool. Z-scores regarding memory declined from 0.97 to −0.45 and −0.4 to −0.56, respectively. The other three players had either equal values, a minimal decrease (e.g., 97 to 96 points), or even a slight increase in their observed scores.

## 4. Discussion

The most important finding of the present study is that SRCs are a common in-competition injury with a highly individual recovery process in professional ice hockey athletes. In training, however, this injury pattern plays a subordinate role, as observed in this cohort. Symptoms of SRC typically dissolve within a few days, while it is nevertheless possible that players miss a substantial number of games due to symptoms lasting up to months.

This seems highly individual, and no obvious risk factor for a prolonged recovery was identified in the observed cohort. A correlation between the initial symptom severity and the observed recovery time, as already published by other authors, appears very plausible. For example, a study from Kowalczyk et al. (2020) found a correlation between initial symptom severity and recovery time in youth athletes [17].

Self-reported symptoms were also reliable predictors, where headache, dizziness, and sensitivity to light seemed to be the most relevant.

NC testing should be carried out, if possible, but it is unclear if NC testing would have the potential for a standalone diagnosis. In the present study cohort, sole NC testing would have underestimated the SRC rate. A study by Gaudet, Konin, and Faust (2021) critically examined the test–retest reliability of this particular NC testing regime and drew concern about the actual reliability of this tool [18]. This seems consistent with the current findings.

The season’s concussion rate was 1.35/1000 AEs. The rate was 5.20/1000 for in-competition matches, and 0/1000 for practices. This is higher than in most previous studies reported (in-competition rate 1.49, 2.49, 7.50; season rate 0.41, 0.50, 0.70, 0.79, 0.97, 1.20, 1.55) [4,5,6]. However, most studies have been conducted on either high school or college athletes, while this study was carried out on professional athletes, where the injury rate in general is higher [19]. Recently, there has been a steady increase in the number of concussion diagnoses. This is commonly related to better recognition and awareness [20].

In addition to possible differences in SRC recognition and awareness, there are relevant regulatory differences between European and North American ice hockey that may lead to higher concussion rates. On the one hand, brawls in Europe, which are a potential risk factor for SRC [21], do not appear to be as much of a problem as in North America [13], which may be associated with a stricter interpretation of the relevant rules and regulations.

However, the comparably high rate of concussions during the season could also be influenced by other factors, such as the different rink sizes and possibly different speed of play. On the other hand, given the fact that the study was conducted for one team during one season, these results should not be overinterpreted, but rather should be seen as requiring further research focusing on ice hockey injuries in professional European leagues.

Four out of five concussed players recovered within a few days but nevertheless missed a mean of three games, which was, on the one hand, due to a fast-paced game schedule and, on the other hand, due to the implemented gradual RTS protocol. Considering the post-concussive vulnerability for a second impact and the increased risk of lower extremity injuries after SRC [2], it is severely advised not to rush athletes to a quick RTS [22].

In accordance with the current consensus of the Concussion in Sports Group, rest or moderate daily physical tasks were suggested to the injured athletes for the first 24 to 48 h after injury, and moderate daily physical tasks were suggested as long as there was no worsening of symptoms [9].

Jildeh et al. (2022) found that, in college and professional athletes, within a 90-day period after sustaining an SRC, the pooled OR for a subsequent lower extremity injury was 3.44 (95% CI, 2.99–4.42), compared with athletes who had not sustained an SRC [23]. In this study, no lower extremity injuries after SRC occurred.

Regarding possible long-term consequences of SRC, it is still controversial if concussions lead to an increased risk of developing neurocognitive disorders later in life. A systematic review by Manley et al. (2017) found that athletes with multiple prior concussions were not at higher risk of suicide, but some may be at risk of diminished or impaired cognitive function. Furthermore, neuroimaging studies showed macrostructural, microstructural, and neurochemical changes in some former athletes [1]. Although a concussion can manifest in a variety of symptoms, it seems likely that there are symptoms that are more specific. Headache, dizziness, and sensitivity to light are very common concussion symptoms, while none of these symptoms were frequently mentioned during baseline testing. Benson et al. (2011) reported that, in 559 cases, the most common concussion symptoms were headache (71%), dizziness (34%), nausea (24%), neck pain (23%), fatigue (22%), and blurred vision (22%) [5]. On the other hand, in the present study, symptoms such as fatigue, trouble falling asleep, and irritability seem to be more common during the exhausting preseason period than during an acute SRC. The fact that athletes already report often multiple symptoms at baseline testing is consistent with previous studies. While the present cohort reported 3.91 ± 5.36 symptoms on average, Cottle et al. (2017) noted athletes reporting 4.0 ± 7.8 symptoms on average at baseline [24].

Additionally, the NC test alone was not decisive enough. In this cohort, only two players showed a substantial decrease in NC function in at least one category. A diagnosis should therefore not be made solely on the basis of self-reported symptoms, NC testing, or neuroimaging, but always in conjunction with a thorough clinical examination, and if necessary, further radiological imaging to exclude more severe injuries [7,25,26].

We acknowledge the limitation of the relatively low number of studied professional athletes in our study; it is especially common in prospective studies on high-level athletes.

Currently, studies with children and adolescents account for over 60% of studies evaluating SRC prevention strategies [9,27].

For ice hockey in general, there is a rather broad body of evidence describing prevention possibilities. Among other things, evidence suggests that rule and policy changes disallowing body checking in youth and adult ice hockey prevent concussions [9,27,28]. Furthermore, personal protective equipment such as mouthguards are seen as useful options for injury prevention [9,27]. In addition, measures to increase awareness as well as clearly comprehensible concussion management strategies are widely recommended by the relevant leagues and clubs. Such measures are considered sensible in order to reduce the overall frequency of SRCs and to improve the safety of players regarding the potential long-term effects of repeated injuries.

In conclusion, our knowledge of the potential long-term consequences and applicable injury prevention strategies for the European professional ice hockey player population is still very incomplete. By conducting our ongoing study, we aim to contribute to addressing this gap and providing a more comprehensive understanding of the injury landscape in professional European ice hockey.

Statistical analysis was partially limited to a descriptive level due to the rather low number of injury cases. However, the descriptive analysis remains valuable, as it allows us to show that circumstances might be different in other levels of play. Accordingly, the results of studies that mainly include nonprofessional athletes should be viewed with caution when attempting to extrapolate these findings to a professional cohort.

Considering both the high number of SRCs already specified before the season as well as the relatively high rate of concussions in the observed season compared with the current literature, the factors that promote the occurrence of ice-hockey-related SRCs, especially under European Games conditions, should be further investigated in future research.

Centralized injury surveillance systems might be useful to gain a better understanding of ice-hockey-specific injuries and to identify some of the relevant risk factors and possible protective measures to prevent serious injuries [29].

Furthermore, this study demonstrates the need for a large-scale collaborative effort between players, coaches, and regulatory and medical staff to better understand SRC in European ice hockey and to optimize its treatment.

## 5. Conclusions

SRC is a common in-competition injury in professional ice hockey and was the main reason for games missed in the present study. The corresponding symptoms and the course of recovery appear to be very individual, as observed in this group of professional athletes. Diagnosis and treatment should therefore not be based solely on (self-reported) symptom scores or individual neurocognitive tests, but always in combination with a detailed clinical (follow-up) examination by a trained physician and, if necessary, additional radiological imaging.

## Figures and Tables

**Table 1 healthcare-11-03153-t001:** Demographic data.

	*n*	Minimum	Maximum	Mean	Std. Deviation
No concussion					
Height in cm		174	191	183.80	5.40
Weight in kg		65	100	84.50	8.15
Age		19	34	26.20	4.92
Games played		2	48	38.45	15.14
Valid (*n*)	20				
Concussion					
Height in cm		176	187	182.60	4.83
Weight in kg		76	92	84.20	6.34
Age		24	33	27.40	3.65
Games played		30	45	38.40	6.88
Valid (*n*)	5				

**Table 2 healthcare-11-03153-t002:** Concussion rate per 1000 AEs.

	Athlete Exposures	Concussions	Concussion Rate ^1^
Games	961	5	5.20
Goalkeepers	52	0	0
Defensemen	330	2	6.06
Forwards	579	3	5.18
Practices	2.735	0	0
Combined	3.696	5	1.35

^1^ Concussions per 1000 athlete exposures.

**Table 3 healthcare-11-03153-t003:** Concussions in comparison with other injuries ^1^ (*n* = 25).

	Total	Mean	Std. Deviation
SRC	5	0.20	0.41
Goalkeepers	0	0	0
Defensemen	2	0.25	0.46
Forwards	3	0.20	0.41
Games missed due to SRC	29	1.16	3.48
Goalkeepers	0	0	0
Defensemen	6	0.75	1.39
Forwards	23	1.53	4.41
Other injuries ^1^	16	0.64	0.70
Goalkeepers	2	1	1.41
Defensemen	6	0.75	0.71
Forwards	8	0.53	0.64
Games missed due to other injury ^1^	24	0.96	2.97
Goalkeepers	0	0	0
Defensemen	14	1.75	4.95
Forwards	10	0.67	1.50

^1^ Lacerations, fractures, strains, and illness.

**Table 4 healthcare-11-03153-t004:** Symptoms at baseline and after concussion.

Symptoms	Concussion (*n* = 5)	Baseline (*n* = 25)
	*n* (%)	*n* (%)
Headache	5 (100)	1 (4)
Dizziness	4 (80)	1 (4)
Sensitivity to light	4 (80)	0 (0)
Feeling mentally foggy	3 (60)	1 (4)
Fatigue	2 (40)	10 (40)
Trouble falling asleep	2 (40)	9 (36)
Difficulty concentrating	2 (40)	4 (16)
Drowsiness	2 (40)	2 (8)
Feeling slowed down	2 (40)	1 (4)
Sensitivity to noise	2 (40)	0 (0)
Nausea	2 (40)	0 (0)
Irritability	1 (20)	6 (24)
Balance problems	1 (20)	2 (8)
Nervousness	1 (20)	1 (4)
Visual problems	1 (20)	1 (4)

Symptoms reported at post-concussion test, in relation to symptoms reported at the baseline testing during the preseason.

**Table 5 healthcare-11-03153-t005:** Individual symptom scores after concussion and changes to the ImPACT test scores.

	Player A	Player B	Player C	Player D	Player E
Symptom score	13	22	5	4	4
Verbal memory ^1^	−6	−1	+4	+11	+5
Visual memory ^1^	−29	0	+1	+11	−21
Visual motor speed ^1^	−1.87	−0.6	+1.97	−2.75	−1.38
Reaction time ^1^	−0.03	−0.05	−0.05	+0.14	+0.05
Impulse control ^1^	−1	+2	−2	+2	−2
Two-factor score memory ^1^	−1.42	−0.05	+0.22	+0.97	−0.58
Two-factor score speed ^1^	−0.29	−0.3	−0.12	−0.49	+0.15
Games missed	3	17	3	3	3

^1^ ‘−’ deterioration to baseline; ‘+’ improvement to baseline.

## Data Availability

Data are held accountable by the team physicians of the regarding team.

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
