# Peer review of "Injury Incidence, Outcomes, and Return to Competition Times after Sports-Related Concussions during One Professional Ice Hockey Season: A Prospective Cohort Study"

_healthcare, 2023, doi:10.3390/healthcare11243153_

Round 1

Reviewer 1 Report (New Reviewer)

Comments and Suggestions for Authors

The manuscript aim is to determine the number of concussions per 1000 Athlete Exposures (AE) diagnosed during one regular season of play in Ice Hockey Players. 

First of all, I would like to congratulate the authors for such an interesting and little studied topic.

Below are some comments that I hope will help to improve the manuscript:

The introduction is well developed and justified.

Materials and method: 

Include in this section that participants signed a prior consent form.

Justification of sample size

Discussion: I would include studies that discuss the possible consequences of brain conxussions not only in the short term, but also in the long term and as a focus of future research.

Author Response

The manuscript aim is to determine the number of concussions per 1000 Athlete Exposures (AE) diagnosed during one regular season of play in Ice Hockey Players. 

First of all, I would like to congratulate the authors for such an interesting and little studied topic.

Below are some comments that I hope will help to improve the manuscript

Thank you very much for this positive review and the helpful comments, which we have implemented as follows:

The introduction is well developed and justified.

Materials and method: 

Include in this section that participants signed a prior consent form.

Justification of sample size

We have now added in detail why the total number of athletes observed and the number of SRC in this cohort appears small compared to other studies outside of professional sport. We have included a statement on the consent of the included athletes at the end of the manuscript in the field provided by the journal.

Discussion: I would include studies that discuss the possible consequences of brain conxussions not only in the short term, but also in the long term and as a focus of future research.

Thanks for this remark. We have added research discussing the possible long term consequences of SRC.

Reviewer 2 Report (New Reviewer)

Comments and Suggestions for Authors

some fragmens should be corrected.
The conclusion sounds very week, and not easy to see why this research is needed. It can be fortified.

The difference between NHL and european league might be a good support to enhance the paper.

Author Response

Some fragmens should be corrected.

fragment sentence. the paragraph needs to be arranged.

fragment sentences.

Thanks for these remarks. This has been adapted.

the conclusion seem quite week compare to the results of the study. It needs to be strengthened.

The conclusion sounds very week, and not easy to see why this research is needed. It can be fortified.

We have tried to re-write the conclusion as well as other parts of the manuscript to better communicate why we think that this research is helpfull for the overall body of evidence regarding SRC in professional ice hockey.

Season and game difference were well compared, however, there might be rule differences between NHL and european league. If there are some differences, it would be good to compare, and discuss possible cause of difference

The difference between NHL and european league might be a good support to enhance the paper.

Thank you for your comment! We have tried to revise and improve the points mentioned accordingly. Therefore, we have rephrased some parts and added further passages on the differences between NHL and european league ice hockey as requested.

future study suggestion may combined with the limitation paragraph because this paragraph is only one sentence and it seems fragment

We have added future study suggestions and have now more clearly stated the limitations of our study.

Reviewer 3 Report (New Reviewer)

Comments and Suggestions for Authors

Hi

I've put my comments in the text

Please be more precise in your description of the protocol and your results.

Author Response

Thank you very much for your thorough review. We have revised the manuscript in line with the comments noted in the text and are convinced that it has benefited greatly from this review.

This data base concern only competition or also the training of these teams?

We have rephrased several parts of the manuscript to make clear that we considered both trainings and competitions.

Before his first training because concussion can occur during training?

Thank you for this comment! We fully agree that an SRC can be acquired during training too, but it was not feasible in this study to test all athletes before the first training session, so we added the passage that testing was done until the first game of the season.

How many days does each stage take? Could you be more specific?

We agree and have now added further information regarding the RTS and presented the steps in more detail. Among other things, it is now clearly stated that "every step lasted at least one day, meaning that athletes could only clear one step each day."

How many players had a concussion before being included in your study?

Thanks a lot for this valuable remark regarding a point we previously missed to mention. We have now clearly stated the exact number of players (n=12) who reported a previous SRC.

Do you have the return to sport times for these 5 players? did any of these players suffer a concussion again, or did they have more injuries than the other non-concussed players?

In order to hinder individual identifiability in view of the fortunately small number of cases of injuries, we have not given an individual number of days lost or exact RTS for each case, but only the median and min./max. number of matches missed. If a more detailed presentation is required, we are willing to provide it.
We have also included a statement that there were no further subsequent injuries after SRC in the cohort observed.

It's better to put this data in the chapter result.

We agree and have rephrased the relevant sentence and moved it to the results section.

It's not clear from your results that concussions cause more unavailability for matches than other injuries (I haven't found any tables relating to these injuries).

It is now clearly stated that "Compared to other reasons for games missed e.g., illness or other injury, concussions were substantial and the main reason for games missed (29) during the season."

Reviewer 4 Report (New Reviewer)

Comments and Suggestions for Authors
Dear Editor,

Thanks a lot for inviting me as the reviewer of the manuscript entitled: "Injury Incidence, Outcomes and Return to Competition Times after Sports Related Concussions during one Professional Ice Hockey Season: A Prospective Cohort Study" in which the authors have presented the number of concussions based on Athlete exposure and reported symptoms in 5 athletes with concussion. I read the manuscript carefully and tried to review it accurately. First, I appreciate the effort put to this job by the authors as they have addressed an interesting topic (SRC) in concussion research; however, there are some concerns about their study mostly in terms of study design, methodology and manuscript drafting.

1- We should accept that there are controversies in definition of concussion; however, there are some criteria which can be considered while conducting a study such as that of 
the 6th International Conference on Concussion in Sport or American Congress of Rehabilitation. Authors have not precised that which diagnostic criteria they have used for diagnosing concussion in the athletes. At some point in the method section, it seems that self-report of symptoms has been considered as the criteria for concussion which should not be. Patients may fulfil the concussion criteria but show no significant symptoms. This might have affected the true incidence of the concussion in the study population.

2- It is recommended that authors address the rational for their sampling method. Why a single team was selected? It may cause selection and sampling bias and affect the generalizability of the findings.

3- Authors have included any athlete with any match or practice duration. I think that may also cause bias as the higher the duration of match or practice, the higher the probability of concussion would be.
4- Authors have defined their own RTS protocol and treated the athletes, accordingly. So, all the durations of sick leave and time to return to sport, would be adjusted by this protocol. Although this protocol considers symptom severity, but it is not a good reflection of  the real effects of severity of symptoms on the duration of sick leave and time to return to sport.
5- Conclusion section is not really a reflection of findings of the present study. For example, authors have mentioned that diagnosis should not be only based on the self reports; however, it seems that they have done the same and moreover, they did not assess diagnostic methods in their study. In addition, does "common" really mean common if 5 out of 25 athletes had concussions and we have not assessed a large sample of athletes.
6- There are also some minor concerns about this manuscript which could be addressed following resolution of the current major ones. 
Finally, I have the impression that the manuscript needs to be re-structured to be more clear for readers in terms of both definitions and scientific writing.

Author Response

Thanks a lot for inviting me as the reviewer of the manuscript entitled: "Injury Incidence, Outcomes and Return to Competition Times after Sports Related Concussions during one Professional Ice Hockey Season: A Prospective Cohort Study" in which the authors have presented the number of concussions based on Athlete exposure and reported symptoms in 5 athletes with concussion. I read the manuscript carefully and tried to review it accurately. First, I appreciate the effort put to this job by the authors as they have addressed an interesting topic (SRC) in concussion research; however, there are some concerns about their study mostly in terms of study design, methodology and manuscript drafting.

Thank you very much for your thorough review of our manuscript. We have revised it in line with your comments and believe that it has clearly benefited from the revision.

1- We should accept that there are controversies in definition of concussion; however, there are some criteria which can be considered while conducting a study such as that of the 6th International Conference on Concussion in Sport or American Congress of Rehabilitation. Authors have not precised that which diagnostic criteria they have used for diagnosing concussion in the athletes. At some point in the method section, it seems that self-report of symptoms has been considered as the criteria for concussion which should not be. Patients may fulfil the concussion criteria but show no significant symptoms. This might have affected the true incidence of the concussion in the study population.

With regard to the definition of SRC, we have referred to the most recent consensus publication of the Concussion in Sports Group, which is now specifically stated and adequately referenced. 

With regard to the specific diagnosis, it is important for us to emphasize that the symptom score was in no way relied upon exclusively, but rather formed a decidedly integral part of the overall assessment, which primarily consisted of a thorough clinical examination (by the team physicians and partly by other medical disciplines), in some cases radiological imaging and the aforementioned impact test. We have tried to formulate this clearly in several places in the manuscript so as not to raise any doubts about an adequate and evidence-based clarification of the SRC cases mentioned.

2 - It is recommended that authors address the rational for their sampling method. Why a single team was selected? It may cause selection and sampling bias and affect the generalizability of the findings.

The decision to observe a single team was made on purely practical grounds. The authors of the study only follow this ice hockey team. We fully understand and appreciate the desire for broader data collection and better evidence in this regard overall - even if we cannot provide this more comprehensively with this study.

Furthermore, we understand the difficulties arising from the overall small number of observed cases, but believe that we have taken every precaution to reduce bias in this respect and to clearly state this limitation.

3- Authors have included any athlete with any match or practice duration. I think that may also cause bias as the higher the duration of match or practice, the higher the probability of concussion would be.

The main findings of this study relate to the overall incidence of SRC in professional ice hockey. We have tried to avoid statements about individual players. We have also gone to great lengths to reduce risk of bias. In our opinion, the overall incidence should not change significantly if the number of players remains constant, regardless of individual ice time. 
As in numerous similar studies, we deliberately calculated athlete exposures (AE) per game, per practice and in accumulation during the whole season in order to reduce bias here as well.
We have only excluded one athlete who did not play at least one game for the respective team during the regular season 2022/2023. Any more generous exclusion would, in our view, lead to an increase in the risk of bias. 

4 - Authors have defined their own RTS protocol and treated the athletes, accordingly. So, all the durations of sick leave and time to return to sport, would be adjusted by this protocol. Although this protocol considers symptom severity, but it is not a good reflection of  the real effects of severity of symptoms on the duration of sick leave and time to return to sport.

We must emphasize that the RTS program is clearly aligned with the most recent evidence-based consensus recommendations of the Concussion in Sports Group, as well as the sport-specific recommendations of the ICE Hockey League. We have now formulated the practical on-site implementation in more detail. We hope that this will contribute to a better understanding.

5 - Conclusion section is not really a reflection of findings of the present study. For example, authors have mentioned that diagnosis should not be only based on the self reports; however, it seems that they have done the same and moreover, they did not assess diagnostic methods in their study. In addition, does "common" really mean common if 5 out of 25 athletes had concussions and we have not assessed a large sample of athletes.

We agree that the conclusion section can be improved and have re-written this paragraph. SRC were the most common injury in the observed cohort and the most common reason for missed matches. We would therefore like to leave the wording as chosen. The limitation of the overall small number of professional ice hockey athletes as well as the correspondingly small number of SRC cases is now stated even more clearly.

6- There are also some minor concerns about this manuscript which could be addressed following resolution of the current major ones. 
Finally, I have the impression that the manuscript needs to be re-structured to be more clear for readers in terms of both definitions and scientific writing.

We would like to thank the reviewer for his/her work and kindly ask to mention the other minor points for further improvement of our manuscript. We believe that the manuscript in the present version has already benefited considerably from the review and that some important points are now much better presented / easier to understand.

Round 2

Reviewer 2 Report (New Reviewer)

Comments and Suggestions for Authors

Thank you for your corrections.

Author Response

Thank you very much for your help in improving this manuscript!

Reviewer 4 Report (New Reviewer)

Comments and Suggestions for Authors

Dear Editor,

Thanks a lot for inviting me as the reviewer for a revision of the present manuscript. I am glad to say that authors have addressed most of my concerns and the interpretation of the results are now more coherent with the methodology and the findings. There are some minor comments that need to be addressed. 

1-It is recommended to check the keywords by MeSH terms in PubMed as it will affect the searchability of the manuscript. It is recommended to use two- or three-word keywords rather than acronyms or single-word terms.

2- In my opinion, it would be better if the authors rephrase the definition of sport-related concussion, by concussion in sport group, in the introduction section instead of direct mentioning. This way it would be more clear.

3- There are some minor style-related errors in the manuscript. For example, the sentence in line 146 is started by "25" and it is not recommended to start a sentence by numbers. I recommend authors to have their manuscript proof-checked by a native scientific writing expert.

Author Response

Thanks a lot for inviting me as the reviewer for a revision of the present manuscript. I am glad to say that authors have addressed most of my concerns and the interpretation of the results are now more coherent with the methodology and the findings. There are some minor comments that need to be addressed. 

Thank you very much for your thorough review and your assistance in improving our manuscript.

1-It is recommended to check the keywords by MeSH terms in PubMed as it will affect the searchability of the manuscript. It is recommended to use two- or three-word keywords rather than acronyms or single-word terms.

We proceeded accordingly and selected the search terms as in similar studies, e.g. 

Pfister T, Pfister K, Hagel B, Ghali WA, Ronksley PE. The incidence of concussion in youth sports: a systematic review and meta-analysis. Br J Sports Med. 2016;50(5):292-297. doi:10.1136/bjsports-2015-094978   Anderson GR, Melugin HP, Stuart MJ. Epidemiology of Injuries in Ice Hockey. Sports Health. 2019;11(6):514-519. doi:10.1177/1941738119849105   Pierpoint LA, Collins C. Epidemiology of Sport-Related Concussion. Clin Sports Med. 2021;40(1):1-18. doi:10.1016/j.csm.2020.08.013  

We have further removed acronyms from the keywords as requested by the reviewer.

2- In my opinion, it would be better if the authors rephrase the definition of sport-related concussion, by concussion in sport group, in the introduction section instead of direct mentioning. This way it would be more clear.

We have rephrased the definition as requested by the reviewer.

3- There are some minor style-related errors in the manuscript. For example, the sentence in line 146 is started by "25" and it is not recommended to start a sentence by numbers. I recommend authors to have their manuscript proof-checked by a native scientific writing expert.

Thank you for this remark. We have modified the beginning of the mentioned sentence as correctly suggested by the reviewer.

This manuscript is a resubmission of an earlier submission. The following is a list of the peer review reports and author responses from that submission.

Round 1

Reviewer 1 Report

Comments and Suggestions for Authors

Comments on the Quality of English Language

Minor copyediting needed for clairty.